# Degradation of Steel Wires in Bimetallic Aluminum–Steel Conductors Exposed to Severe Corrosion Conditions

Alan Rondineau [1,2,*], Laurent Gaillet [1], Lamine Dieng [1] and Sébastien Langlois [2,*]

1   Laboratoire de Structures Métalliques et à Câbles (SMC), Département Matériaux et Structures (MAST), Université Gustave Eiffel, 44344 Bouguenais, France
2   Département de Génie Civil, Université de Sherbrooke, Sherbrooke, QC J1K 2R1, Canada
*   Correspondence: alan.rondineau@univ-eiffel.fr (A.R.); sebastien.langlois@usherbrooke.ca (S.L.); Tel.: +33-771590440 (A.R.)

**Abstract:** High-voltage electrical cables are prone to saline corrosion, mostly in coastal environments. Steel wires are a crucial element in withstanding the mechanical solicitations of commonly used aluminum conductor steel reinforced (ACSR) cables. An experimental accelerated corrosion test was made, using salt spray tests on greased and ungreased ACSR cables and individual galvanized steel wires. The corrosion mechanism occurring on the specimens was observed by optical microscopy for several durations of corrosion, to determine the evolution of the galvanic layer and steel substrate degradation. This study was completed by an SEM (Scanning Electron Microscopy) and Raman spectroscopy analysis to characterize the corrosion products occurring on the galvanized steel wires. An estimation of the evolution of the mean zinc thickness loss is also given, for each type of specimen. It is shown that the loss rate of the zinc layer is significantly reduced by the presence of aluminum layers around the steel wires and by the effect of the grease. Tensile tests were made on the exposed galvanized steel wires which led to fracture surface observations to assess the effect of corrosion on the evolution of mechanical properties.

**Keywords:** cables; corrosion; galvanized steel; ACSR conductors

## 1. Introduction

Civil engineering conductors are mainly subjected to two different damage mechanisms, namely fretting fatigue, and corrosion. These two mechanisms can significantly affect the cable lifetime and jointly, can significantly increase the total damage to wires [1]. Nowadays, power transmission system operators are struggling to have a good estimation of the remaining lifetime of power lines in use, since the corrosion behavior on high-voltage electrical conductors is difficult to evaluate.

Aluminum conductor steel reinforced (ACSR) conductors are commonly used in the world because of their high mechanical stiffness and strength properties (steel part) and good electrical conductivity (aluminum part). Those multilayer strand wires are made of high-strength steel and aluminum 1350 H19. The steel wires are typically protected from corrosion with a zinc galvanization layer. This type of cable has a highly complex corrosion behavior given its multi-material characteristics (aluminum, steel, and zinc) and its multilayer stranded geometry.

C. O. Ujah and al. [2] investigated the performances of high-temperature low sag (HTLS) cables with corrosion and revealed that ACSR conductors are very sensitive to corrosion, more than other conductors, because of the zinc layers around steel wires being corroded in the ACSR conductors. The total corrosion of the zinc layers leads to an increase in the corrosion of the steel wires and a total breakage of the cable.

Forrest and al. [3] studied the mean rate of deterioration of bimetallic conductors in coastal environments. Their work highlights the galvanic corrosion occurring between steel and inner aluminum wires in marine environments, in which inner aluminum wires may

corrode much faster than outer aluminum wires. However, the steel core was reported to be uncorroded.

E. Håkansson and al. [4] have studied the galvanic corrosion mechanism occurring in high-voltage cables such as ACSR conductors. Their studies revealed the dependence on the temperature at which the metal will be more noble in steel, zinc, and aluminum. They found that aluminum is less noble than zinc and steel at 85 °C, whereas aluminum is more noble than zinc at atmospheric temperatures. However, the causes of metals' nobility have not been investigated.

Other researchers such as Zhang and al. [5], Schwabe and al. [6], and Lyon and al. [7] have made many investigations into the corrosion of galvanized steel wires from high-voltage conductors using accelerated NaCl corrosion tests. Their work and analyses showed the mechanisms of corrosion occurring in galvanized steel ACSR wires, such as a mix of pitting, galvanic, and crevice corrosion. However, the effect of such corrosion damages on the galvanized steel mechanical properties was not investigated.

Lequien and al. [8] studied ACSR corroded greased cables exposed for 60 years in a mixed rural and urban environment and revealed a degradation of the grease and a reduction in the zinc layer thickness due to its oxidation. Moreover, a degradation representation has been made for a galvanized steel wire, such as a uniform degradation process of the zinc layer, forming a zinc oxide layer ZnO on all of the surface layers. With the rate of corrosion on ACSR conductors being generally very slow, the samples studied did not suffer a significant loss of mechanical strength. In that sense, it is important to complete this type of study with accelerated tests.

The corrosion acceleration process is inspired by both the ISO 9227 [9] and ISO 16701 [10] standards. Considering the ISO 16701 standard, samples are exposed to alternating chloride vaporization. However, the chloride concentration and the pH level are chosen according to ISO 9227, which proposes a classic constant humidity salt spray test. Accordingly, the corrosion solution is made with a concentration of $50 \pm 5$ g/L of salt, to ensure a very severe corrosion process, which is needed to obtain significant corrosion damage for ACSR conductors in a reasonable time. The humidity of the test is cycled to increase the severity of the accelerated corrosion test.

The impact of salt spray tests has been studied by N. LeBozec and al. [11] in comparison with other automotive corrosion tests. The ISO 9227 standard [9] has been studied in comparison with other automotive accelerated corrosion test standards, such as GM9540P or VCS1027, and showed that the ISO 9227 standard may not be realistic in terms of the salt concentration and the results of corrosion for basic materials. The study by LeBozec and al. also highlighted that the corrosion test performance is highly dependent on the test parameters.

This study deals with the type and extent of corrosion damage that occurs in ACSR conductors when subjected to a cyclic salt spray test. Its main objective is to evaluate the effect of the cable geometry and grease on the extent and progress of the rate of corrosion damage to steel wires and galvanization, when compared to zinc and steel reference plates and individual galvanized wires. Firstly, the accelerated corrosion test will be introduced, then the microscopic observations made of the corroded steel samples will be shown. Then, the mass loss measurements of the steel, aluminum, and zinc coupons are presented, and the zinc thickness losses from the zinc coupons and the individual, greased, and ungreased steel wires are studied. The EDS (Energy Dispersive X-ray Spectroscopy) and Raman spectroscopy studies will be presented, revealing the corrosion products occurring on the steel wires. Finally, the results obtained from the mechanical tensile tests of the corroded steel wires will be presented.

To the best of the authors' knowledge, this is the first attempt to directly compare corrosion damage and its effect on the mechanical strength of individual galvanized steel and aluminum wires, and greased and ungreased conductors. This paper also shows a comparison between zinc, aluminum, and steel reference plates with real overhead line conductors. This contributes to a much better understanding and evaluation of the effect of geometry and grease on the corrosion of ACSR cables.

## 2. Materials and Methods

### 2.1. Test Specimens

This work presents the study of greased and ungreased ACSR conductors 1350 series, steel wires, and steel and zinc coupons. The grease used in conductors is OCG 4500 grease, a non-volatile grease insoluble in water, with a melting point higher than 220 °C. This grease is located within the cable, all around the steel wires, up to the first inner aluminum strand. The ACSR conductors and wires were provided by SolidAl company, and the steel and zinc coupons were provided by the Goodfellow Inc. company (Delson, QC, Canada).

The ACSR conductor diameter is 19.6 mm, and the wire diameter is 2.8 mm. This conductor is made up of four different layers with 37 round wires. The center of the cable is made of seven steel strands (one core wire and one layer of six strands). The two outer layers are made of aluminum and have, respectively, 12 and 18 aluminum strands. The ACSR conductor cross section is presented in Figure 1. Gray-colored areas represent galvanized steel wires, and white-colored areas represent aluminum wires.

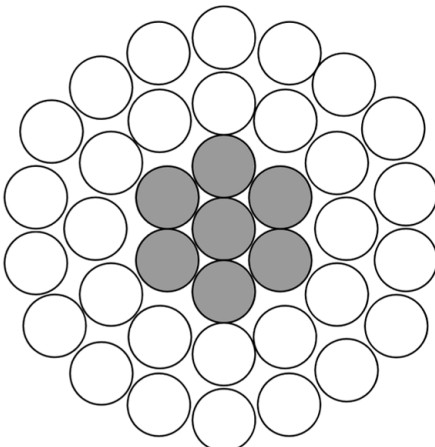

**Figure 1.** Cross section of the ACSR conductor.

The following specimens were tested to assess the corrosion on conductors, ACSR cables, individual steel wires, and steel and zinc coupons. The chemical composition of the steel constituting the cables and individual wires is presented in Table 1.

**Table 1.** Chemical composition of steel wires.

| C (Carbon) | Mn (Manganese) | Si (Silicon) | S (Sulfur) | P (Phosphor) |
| --- | --- | --- | --- | --- |
| 0.80% | 0.52% | 0.23% | 0.018% | 0.017% |

The steel and zinc coupons were placed in the environmental chamber to follow the evolution of metallic corrosion through their macroscopic aspects, according to the ISO 9226-2012 standard [12]. The contour of each coupon was protected by a varnish, preventing any corrosion occurrence at the edges of each coupon.

The zinc coupons were made of more than 99.95% zinc [12]. The steel wires were made of 0.8% carbon, higher than the steel coupons which were 0.041% ± 0.003% carbon.

Both the individual wires and the ACSR cables were made by the same manufacturer, and all the individual steel wires have the same characteristics as the wires constituting the strands in cables. During the preparation, each extremity of the cable was tightened by a plastic clamp to keep the cables stranded.

Then, the greased and ungreased cables and wire samples were sealed with a protective gel at both extremities to avoid corrosion occurring at the ends of the conductors.

The wires and cables from the corrosion test were cut into smaller samples of 10 mm ± 2 mm close to the extremity and outside of the protected extremities. The re-

maining samples were used for further analysis, such as mechanical tests. The samples were embedded in phenolic metallographic coating resin for polishing. For each sample, a cross and longitudinal section were studied and polished to a polishing grain size of 0.03 μm for observation.

### 2.2. Accelerated Corrosion Test Parameters

An accelerated corrosion test was carried out on these specimens, exposing them to a cyclic corrosion salt spray.

The corrosion chamber could control both the temperature and relative humidity parameters for specific durations and contained four different batches of greased and ungreased cables, steel wires, and coupons (steel, and zinc). The samples are placed respecting a minimum distance of 20 mm between each specimen, referring to the ISO 7384 standard [13]. All of the samples were positioned as seen in Figure 2.

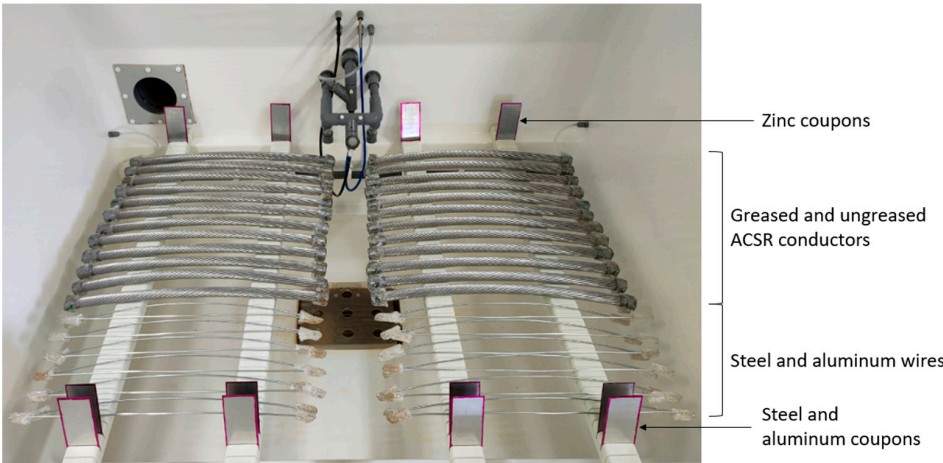

**Figure 2.** Wire and cable arrangement in the corrosive chamber.

The corrosion test was made up of repeating cycles of 84 h, with a pH value set at 6.5–7.2, at a constant temperature of 50 °C. The RH evolution and cycles are presented in Figure 3.

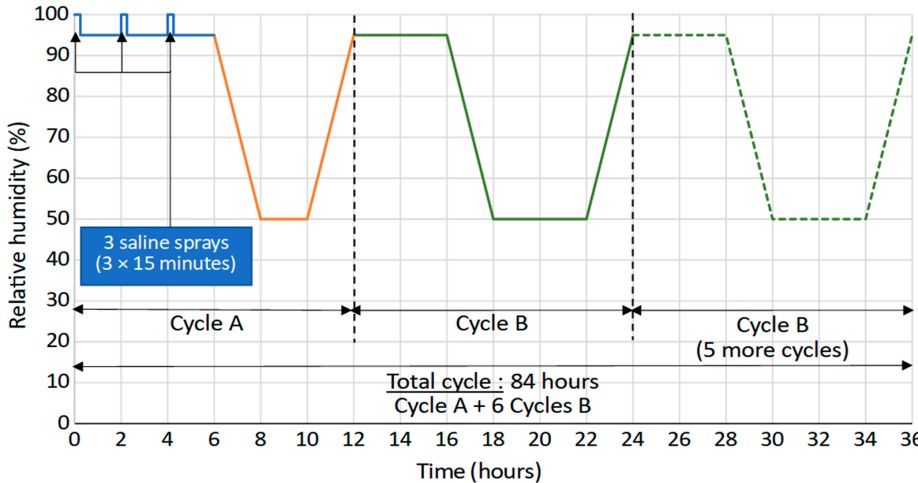

**Figure 3.** Relative humidity and cycles of the corrosion test.

The following test samples were subjected to the accelerated corrosion procedure: reference plates made of two materials (steel, and zinc), galvanized steel wires, and greased and ungreased ACSR conductors. Structural observations and Raman analysis were made

on the test samples and were compared to reveal the impact of the geometrical structure and the presence of grease on the corrosion process.

Each batch was attributed a corrosion duration. A total of seven batches of wire cable and coupon samples were used for this test, for an overall test duration of 336 days. Six durations of corrosion were set for the accelerated corrosion test: 42, 84, 168, 210, 252, and 336 days.

Once the steel samples were prepared for observations, the corrosion products were analyzed through SEM observations and EDS analyses, carried out at low vacuum considering the low electrical conductivity of the corrosion products. The Raman spectroscopy analyses were then performed with a micro-spectrometer and a confocal Olympus BX41 microscope. The wavelength of the HeNe laser source used in the analyses was 532 nm. The software used for database acquisition and processing was LabSpec 6.

The tensile tests were performed in an MTS testing machine. A constant displacement of 6.0 mm/min was set up for the test until the steel wires broke. A 25 mm extensometer ($-10\%$ to $+50\%$ of deformation) was placed on the wire to measure the elongation and calculate the deformation (Figure 4), for further comparisons between the different corroded wires.

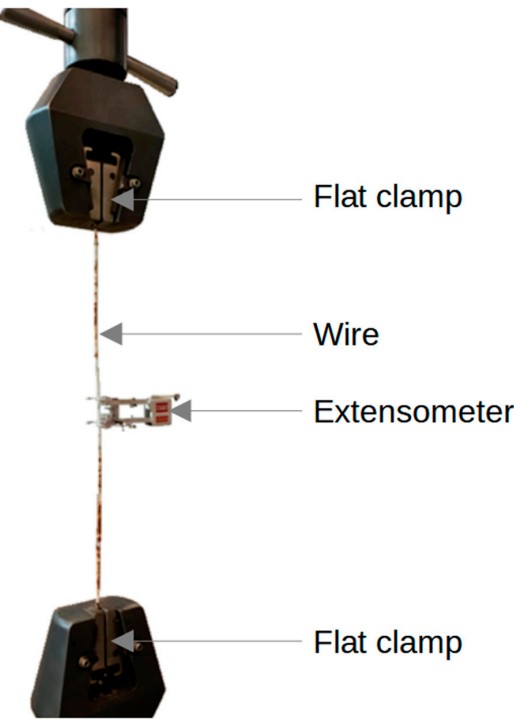

**Figure 4.** Tensile test bench made at the University of Sherbrooke.

The total length of the specimen was set between 350 and 400 mm, and flat clamps were used at both extremities, with a clamping end at 50 mm. The loading bench used for the tests had a total capacity of 1650 kN, with a load cell of 45 kN for the steel wires.

## 3. Results and Discussion

### 3.1. Microstructural Observations

Test specimens were first observed with a binocular microscope. The first step consisted of measuring the corrosion layers on the samples, the local corrosion geometrical dimensions, and the corrosion density at the surface of the wire. For the steel wire, the galvanization layer was also observed to determine which intermetallic layers had been corroded.

The zinc layer on the steel substrate was composed of four intermetallic layers. From the steel substrate to the external surface of the wire, the following intermetallic layers

were observed: gamma, delta, zeta, and eta. These four intermetallic layers were composed of iron and zinc. The further away the intermetallic layer from the substrate and close to the external surface, the higher the zinc content and the lower the iron content. The intermetallic layers are represented in Figure 5.

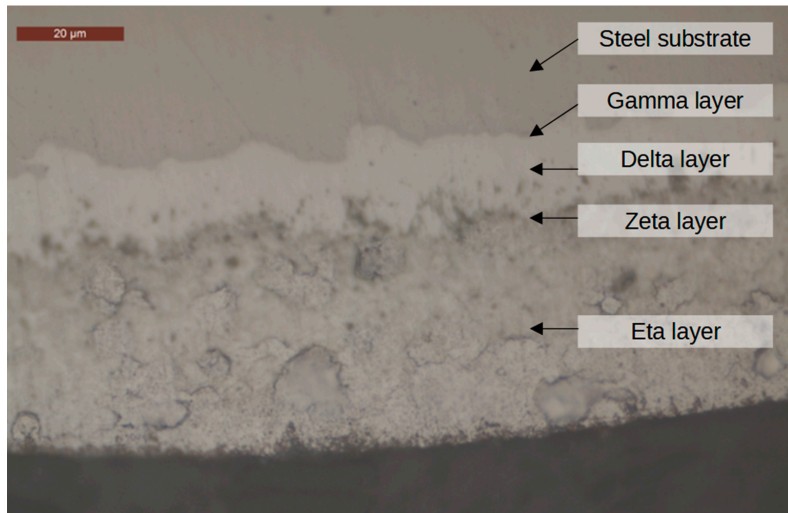

**Figure 5.** Intermetallic layers in the steel substrate.

Figures 6–8 represent the macroscopic degradation observed for an individual galvanized steel wire, an ungreased galvanized steel wire, and a greased galvanized steel wire, respectively.

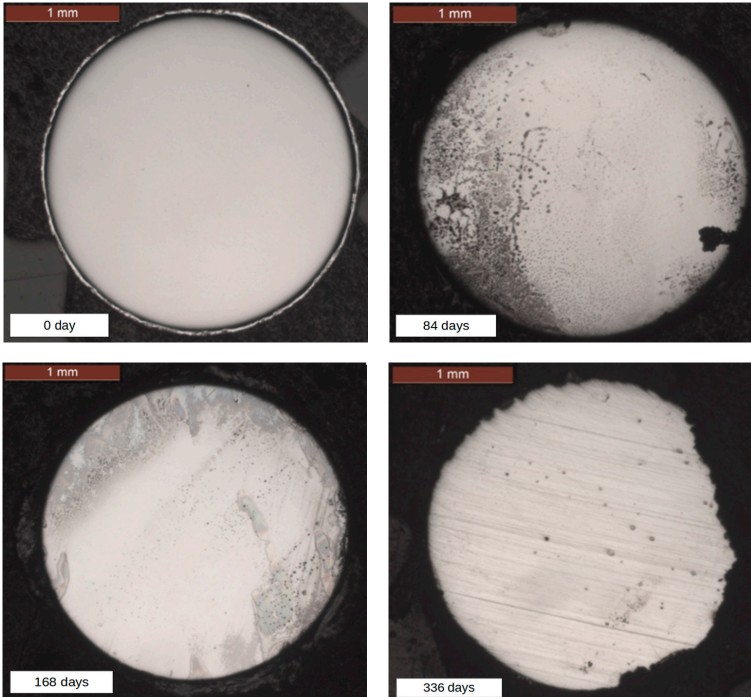

**Figure 6.** Transverse cross section of a corroded individual steel wire.

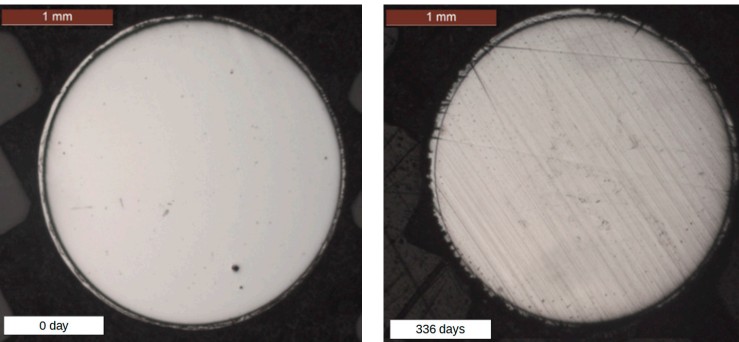

**Figure 7.** Transverse cross section of a corroded steel wire in an ungreased cable.

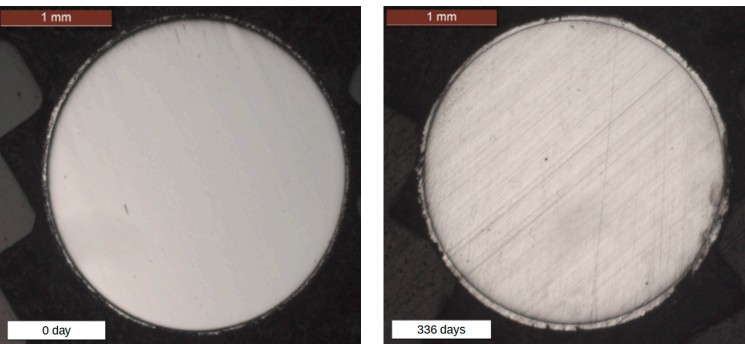

**Figure 8.** Transverse cross section of a corroded steel wire in a greased cable.

At 84 days, local corrosion had occurred on the surface of individual steel wires, despite the presence of the remaining zinc layer (Figure 6). The galvanic protection was completely corroded on the individual steel wires after 168 days. From 168 days, the steel wire was totally exposed to saline corrosion, showing severe corrosion damage. This direct exposure of the steel substrate led to uniform corrosion on all the surfaces, generally localized on the side more exposed to the saline solution. At 336 days, the steel substrate was significantly corroded, leading to an important reduction in the initial diameter.

From 168 days, the remaining zinc was only measured on the greased and ungreased steel wires since the galvanization on the individual wires was totally corroded. The mean zinc values obtained at 336 days for the ungreased and greased steel wires were calculated at 9 µm and 33 µm, respectively, corresponding to 1.78% and 0.94% respective loss of the total diameter; a maximum value was measured of 43 µm and 58 µm (0.59% and 0.07% loss of the total diameter), respectively.

Figure 9 represents the observations made at 252 days of corrosion. At this stage, uniform corrosion occurs around the individual steel wire (Figure 9a). The zinc layer present on the ungreased steel wire (Figure 9b) showed some pitting corrosion on the surface, but not reaching the steel substrate. The steel wires from the greased cable (Figure 9c) remained globally intact.

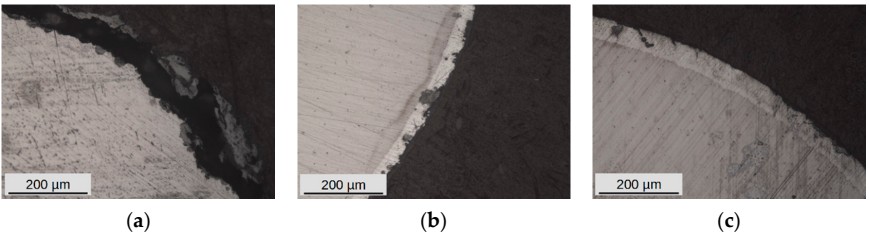

**Figure 9.** Local zinc loss at 252 days of corrosion: (**a**) Individual steel wire, (**b**) ungreased steel wire, (**c**) greased steel wire.

### 3.2. Mass and Thickness Loss

The first results obtained from the coupons gave us information about the severity of the corrosion test. The corrosion products which had formed on the metallic coupons were removed following the ISO 8407 standard [14].

The uncorroded coupons were weighed to measure the mass loss. These data were compared with the ISO 12944-2 [15] standard to determine the corrosivity category of the corrosion test, for each material.

The mass loss results are presented in Table 2.

**Table 2.** Mass loss on corroded coupon samples compared with the CX corrosivity category.

| Metallic Coupon | Aluminum | Steel | Zinc |
|---|---|---|---|
| Mass loss per unit surface (g/m$^2$) | 343 | 8540 | 2042 |
| CX category | >27 | >1500 | >60 |

Each of those results was compared with the data from the ISO 9223 standard [16], attributing a corrosivity category according to the mass loss per unit surface area for the steel and zinc coupons. By comparison, it was noted that all of the metal corrosion rates were in the extreme corrosion category CX, corresponding to high salinity and extreme humidity.

The evolution of the mean galvanization thickness loss for a zinc coupon, an individual wire, and the wires extracted from greased and ungreased ACSR conductors is represented in Figure 10. It can be observed that the zinc loss was significantly higher for a zinc coupon than any of the studied galvanized steel wires. This first result can be explained by the difference in material constitution between the zinc coupon and the intermetallic layers of the steel wires. However, the higher corrosion loss of zinc might also be explained, in part, by the difference in shapes between the flat coupon and the wire. Furthermore, after 168 days, the difference between the zinc coupon and the individual steel wire further increases naturally because the galvanized layer of the individual steel wire has been completely removed at this point.

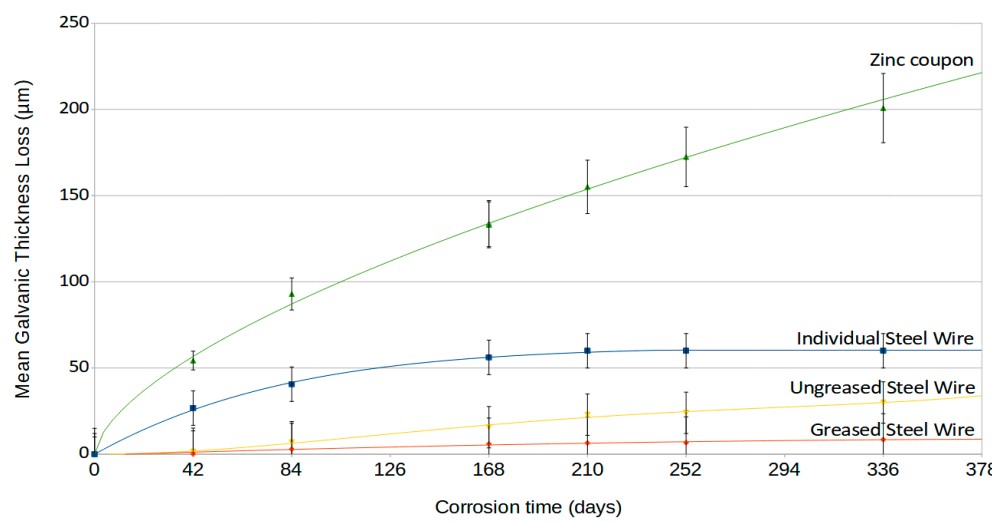

**Figure 10.** Mean zinc thickness loss on galvanized steel wires.

The zinc loss from an individual galvanized steel wire is higher than that for the conductor steel wires. This result is due to the sheltering effect of the two aluminum layers covering the steel wires. The galvanization layer is less corroded for greased steel wires in comparison with the corrosion observations made on individual wires and ungreased wires.

The sheltering effect has already been highlighted by Forrest and al. [3], in which corroded bimetallic conductors were shown to be in a good state despite severe corrosion

of external aluminum wires. These observations fit with the current results, in which steel wires are less corroded than individual steel wires.

As mentioned earlier, the galvanic layer protection of individual steel wires is totally corroded after 168 days of corrosion. The tendency curves obtained from the galvanization layer loss are used to estimate the time at which the zinc layer will be totally corroded for an average zinc layer thickness set at 60 μm. According to the results obtained on ungreased and greased cables, the estimation of total corrosion of galvanic layers on internal steel wires was calculated to be 726 days for ungreased wires, and 11,900 days (around 33 years) for greased wires in this severe environment. This assumes that the grease within the greased cables does not lose any of its properties. The comparison with a real environment is difficult to make at this point, however, these results offer a first relative evaluation of the loss of the galvanized layer from steel wires in different geometrics and materials.

### 3.3. Analysis of Corrosion Products

### 3.3.1. EDS Results

The steel wires were first analyzed in EDS using scanning electron microscopy (SEM) to obtain qualitative information about the chemical elements present on the surface of the steel wires. Additionally, Raman spectroscopy analysis was performed on the same wire samples, to fully identify and quantify the corrosion products.

For comparison, both the uncorroded and corroded states of the steel wires were investigated. This analysis was carried out on an uncorroded steel wire, as presented in Figure 11a, for which the steel substrate and the galvanic layer were analyzed, as shown in Figure 11b,c.

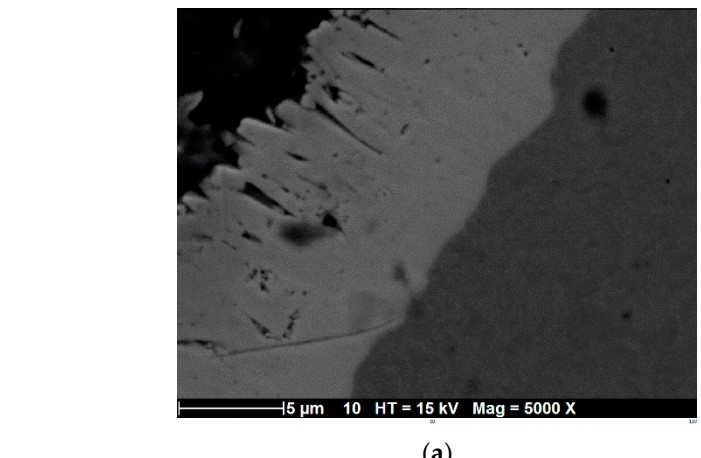

**(a)**

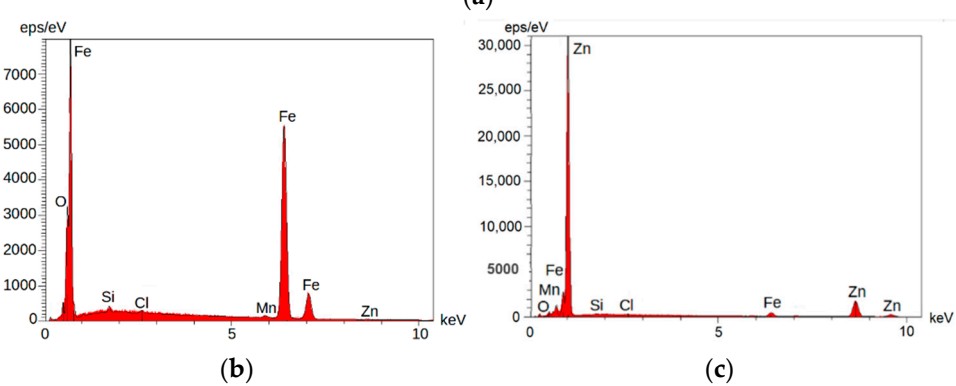

**(b)**          **(c)**

**Figure 11.** Spectroscopy analysis for an uncorroded steel wire: (**a**) Microscopic view; (**b**) steel substrate; and (**c**) galvanization layer.

The EDS analysis of the corroded steel wires revealed the presence of oxygen and iron in the galvanization layer. However, very few elements of chloride and oxygen were

revealed in the substrate and the zinc layer. Other analyses were made for the corroded steel wires, all the galvanic layers, and the steel substrate. Figure 12 presents a cartographic analysis of a galvanic layer from an ungreased corroded steel wire cable after 210 days. The analyses show the Fe, Cl, O, and Zn rates present on the surface.

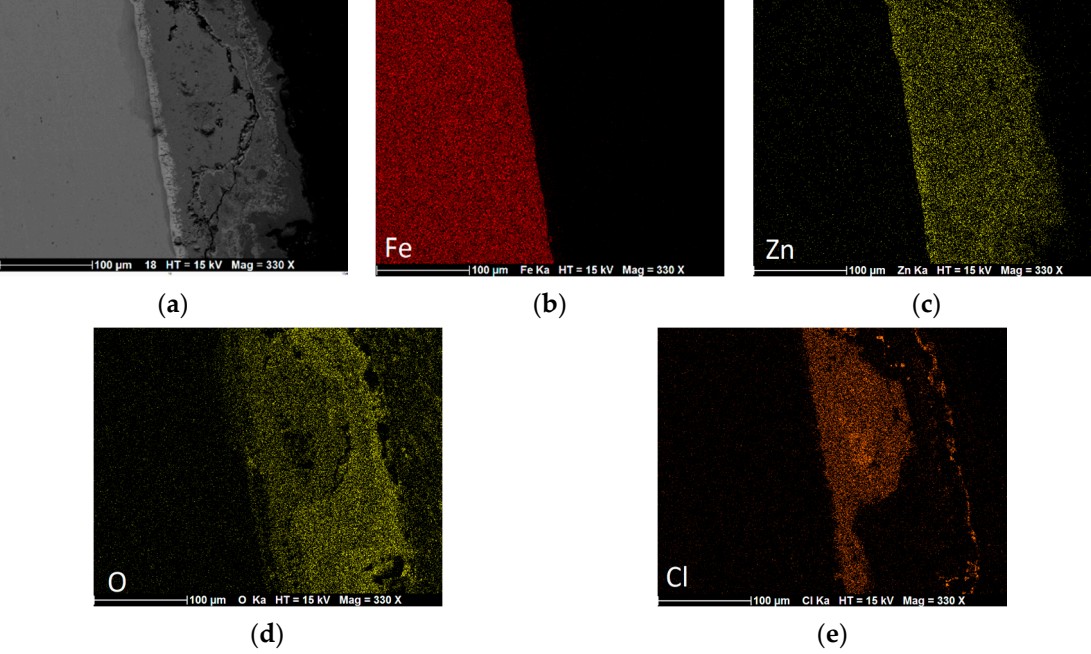

**Figure 12.** EDS cartography of a 210-day corroded steel wire: (**a**) Microscopic view; (**b**) iron levels; (**c**) zinc levels; (**d**) oxygen levels; and (**e**) chloride levels.

Those cartographies reveal an important concentration of chloride (Figure 12c) with oxygen (Figure 12d) in the internal galvanic layers, from the substrate to a cracked galvanic layer.

3.3.2. Raman Spectroscopy Results

Raman spectroscopy can give us information about the nature of the corrosion products formed in the zinc layer and steel substrate. Both analyses were carried out on longitudinal/transverse cross sections, and on the surface of the corroded wires, to get a complete analysis of all the products formed on each galvanic layer and steel surface.

In Figure 13, Raman spectrometry reveals the presence of hydrozincite ($Zn_5(CO_3)_2(OH)_6$), through the peak at 1063 cm$^{-1}$ on the curve (1). This hydrozincite product was also observed through Raman analysis by M. Carbucicchio and al. [17] on galvanized steel.

Figure 13 also reveals the presence of simonkolleite ($Zn_5(OH)_8Cl_2 \cdot H_2O$), from the peaks obtained at 253 cm$^{-1}$, 393 cm$^{-1}$, and 1597 cm$^{-1}$, situated between steel and the external corrosion products.

Those products appear with a clear white aspect. Raman spectroscopy analysis shows the presence of zincite (ZnO) (Figure 13) on curve (1), from the peaks at 440 cm$^{-1}$ and 1074 cm$^{-1}$ (M. Chandra Sekhar and al. [18]), through a macroscopic orange aspect on the galvanic surface.

The last analysis (Figure 13) reveals the zinc ferrite ($ZnFe_2O_4$) product on the surface of the wire, with characteristic peaks at 306 cm$^{-1}$, 545 cm$^{-1}$, and 676 cm$^{-1}$ on curve (2). Zinc ferrite was highlighted by B. Paz and al. [19] while analyzing the constitution of zinc galvanic layers.

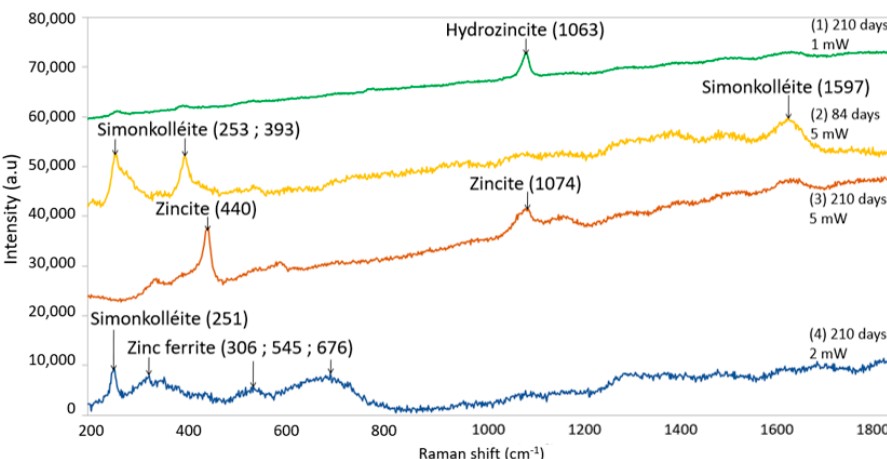

**Figure 13.** Four examples of Raman spectroscopy analysis of a corroded steel wire surface.

Other corrosion tests were performed by Vera and al. [20] on galvanized steel in a corrosive marine environment for further Raman spectroscopy observations. Their analyses revealed the presence of simonkolleite ($Zn_5(OH)_8Cl_2·H_2O$) and zincite (ZnO) on the surface of the galvanization layer.

All of the corrosion products observed through Raman spectroscopy match with pollutants revealed in the literature along the galvanization layer of the corroded steel wires. These SEM and Raman spectroscopy observations led to the same pollutant results for individual and steel wire cables. However, despite the high corrosivity of the accelerated corrosion test, the corrosion products obtained on both the zinc layer surface and the steel substrate fit with the analysis results obtained from corroded galvanized steel by Vera and al. [20], which validates the corrosion test.

*3.4. Mechanical Tension Tests*

3.4.1. Mechanical Test Results

Corroded steel wires were submitted for tensile testing to assess the change in mechanical properties after corrosion. Individual steel wires and steel wires from greased and ungreased conductors for each corrosion time were evaluated.

For individual wires, some specimens at 168 days and higher showed a significant reduction in mechanical strength (Figure 14). This corrosion time corresponds to the moment at which the zinc layer was completely corroded and the steel substrate was consequently corroding.

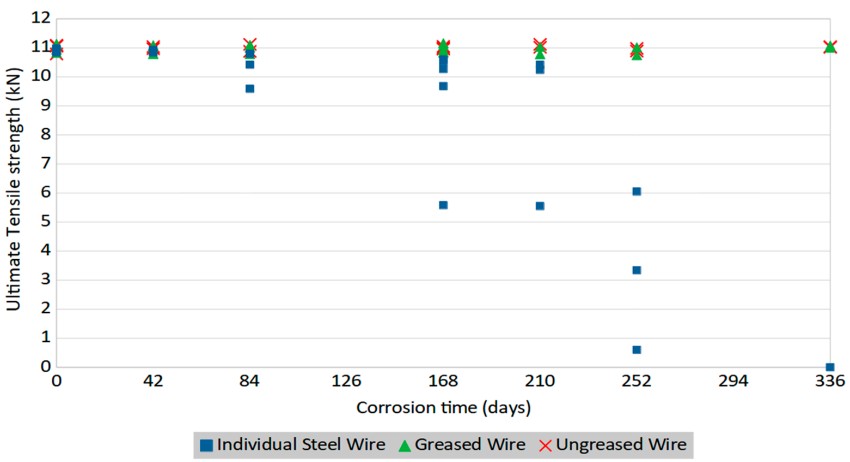

**Figure 14.** Ultimate tensile strength of corroded steel wires for different exposure times.

From Figure 14, it can be seen that no significant change in the UTS values was observed for all corrosion durations for the steel wires extracted from greased and ungreased conductors. This result can be explained through the protective effect of the zinc layer around the steel wire, which is still present on ungreased and greased steel wires, despite the beginning of pitting corrosion observed at 252 and 336 days for ungreased and greased wires, respectively. Moreover, steel wires are geometrically protected by aluminum wires positioned all around the steel wires and directly exposed to the saline solution. Overall, the measured UTS for all uncorroded or lightly corroded wires was very close to the expected UTS value of 11.0 kN.

In Figure 15, the tensile strength deformation evolution is represented for uncorroded steel wires and corroded wires at 168 days. For all samples, except for individual wire at 168 days of corrosion, an elastoplastic transition occurred at around 9.3 kN, and a final rupture strength occurred at around 11 kN with a deformation between 6% and 7%. However, the individual wire at 168 days of corrosion had a sudden rupture at 5.58 kN and 0.45% of deformation. From a visual observation of this specimen, a severe zone of corrosion could be observed. The other part of the wire presented homogeneous corrosion. The reduction in the deformation at rupture is consistent with the tensile strength tests made by S. Nakamura and al. [21] and performed on corroded galvanized steel wires for bridge cables that showed an important loss of ductility for severely corroded wires.

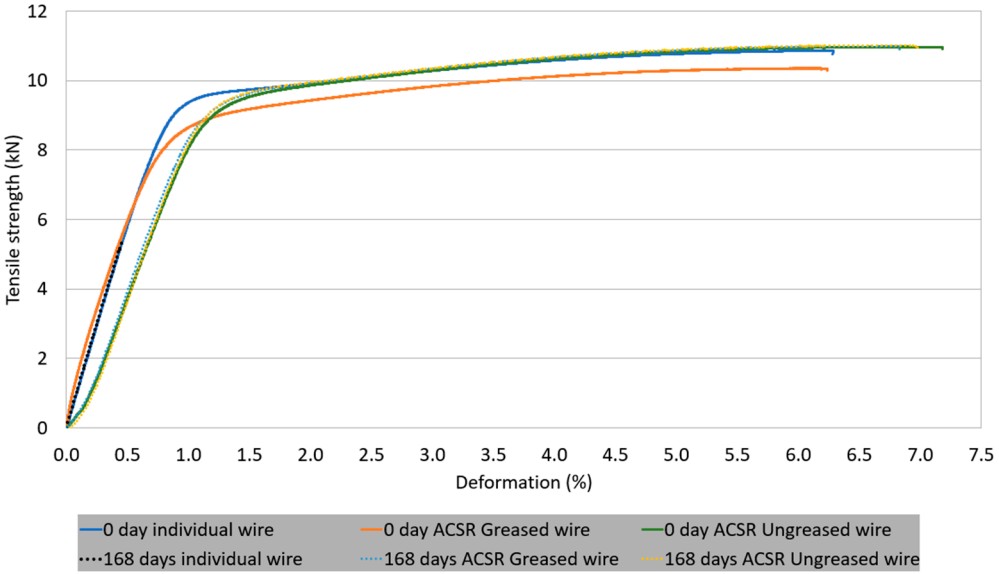

**Figure 15.** Tensile strength deformation evolution on corroded steel wires.

Other specimens of individual wires at 168 and 210 days showed a much more moderate loss of tensile strength. This is in line with the visual observations that show no significant concentration of steel corrosion products in these samples. In that sense, the development of corrosion on galvanized steel shows an important variability from one sample to another, this being certainly related to the nonuniform loss of the zinc layer on the wires, as highlighted in the preceding section §3 concerning microstructural observations. However, as seen in Figure 15, there is a clear trend that the strength of individual wires starts to be significantly affected by corrosion between 168 and 252 days. The test could not be made on the individual steel wires corroded at 336 days since they were very severely corroded and fractured without applied stress during the accelerated corrosion test. The UTS deformation results of the most severely corroded individual wires are presented in Table 3. For the three tests that showed important strength losses at 168, 210, and 252 days of corrosion, a second test was performed excluding the most severe corrosion zone.

**Table 3.** UTS—Total deformation of corroded individual wires.

| Specimen Surface | 168 Days Test | 210 Days Test | 252 Days Test |
|---|---|---|---|
| Including a highly corroded area | 5.58 kN—0.45% | 5.55 kN—0.44% | 3.34 kN—0.16% |
| Excluding the highly corroded area | 10.83 kN—5.79% | 9.86 kN—1.80% | 6.92 kN—0.56% |

The first tests on these three specimens showed a huge decrease in the strength and elongation values. The total elongation measured with the local steel-corroded zone is at 0.16%–0.45%, with a UTS value of 3.34 kN–5.58 kN. For the second test performed on these wires, the UTS and final elongation values were higher than the first tests on steel-corroded zones. However, the UTS and elongation were generally lower than for uncorroded wires, due to some local pitting corrosion areas occurring on the wire. More specifically, the UTS value at 168 days is more or less the same as an uncorroded wire, however, UTS values are lower for 210 days and 252 days. This result can be explained by the microscopic observations, in which pitting corrosion is seen reaching the steel substrate through the zinc layer at 252 days.

### 3.4.2. Tensile Test Fracture Observations

Surface fracture observations were made on the uncorroded and corroded steel wires to discern the effect of corrosion severity on the microstructure.

Figure 16 represent the fracture surface observed in a wire and its conjugate on the other wire side, for a 252-day steel wire from a corroded greased conductor. Figure 17 represent the fracture surface observed in a wire and its conjugate on the other wire side, for a 168-day individual steel wire. The 252-day wire from a greased conductor represents a low-corroded steel wire, whereas the 168-day individual wire represents an intermediate and highly corroded steel wire.

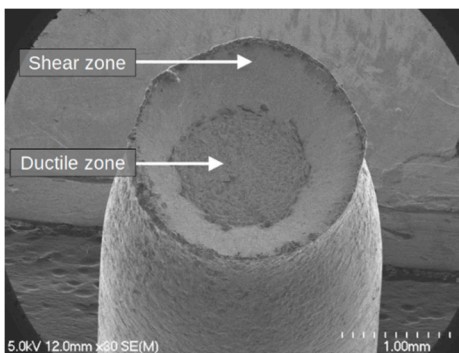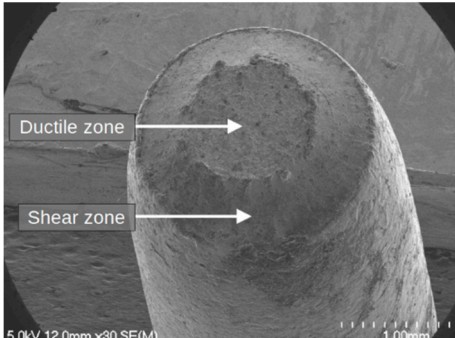

**Figure 16.** Tensile test fracture surface of a low-corroded steel wire (252 days, wire from greased cable, UTS = 10.89 kN, total deformation = 6.10%).

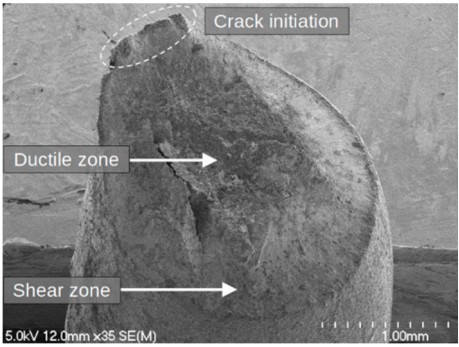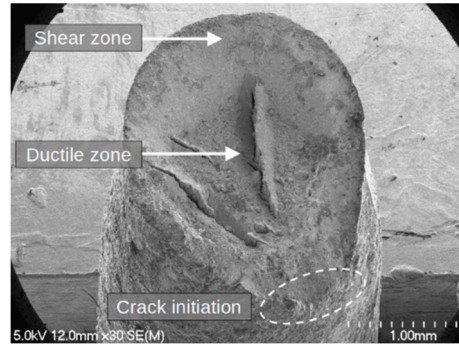

**Figure 17.** Tensile test fracture surface of an intermediate corroded steel wire (168 days, individual steel wire, UTS = 10.53 kN, total deformation = 3.63%).

The fracture surface obtained for the 252-day steel wire in Figure 16, corresponds to a cup-cone shape, representative of a ductile fracture. This shape has been observed by X. Zheng and al. [22] in mildly corroded steel wires, characterized by a fibrous breakage at the center, and a shear lip around the wire. Rou Li and al. [23] have also observed these fracture surfaces during tensile tests on galvanized steel wires. Ductile zones can be observed in Figure 16 at the microscopic level for the fracture surfaces which confirms the ductile behavior of the low-corroded wires

In Figure 17, at 168 days, ductility is decreasing and a shorter crack propagation length with final shear lips is highlighted.

For the 168-day corroded individual steel wire, the UTS was measured at 10.53 kN, for a total deformation of 3.63%. The rupture occurred at a lower total deformation value than the nominal deformation value measured for a low-corroded or uncorroded steel wire. Considering the specific shape of this fracture, the rupture behavior of the corroded steel wire became more brittle once the galvanization layer had completely corroded.

## 4. Conclusions

In this study, different types of specimens, such as steel wires extracted from greased and ungreased ACSR conductors, individual steel wires, and steel and zinc coupons, were studied and exposed to an accelerated corrosion test lasting 336 days. The coupons classified the test severity as an extreme corrosion area, according to the ISO 9223 standard [16]. The galvanized steel wires were observed and analyzed using several techniques. At 168 days, the galvanization layers on the individual wires were completely corroded and severe corrosion of the steel substrate was highlighted. At 252 days, the ungreased steel wire showed pitting corrosion zones in the intermetallic layer, while greased steel wires remained globally intact. At 336 days, the density of the pits had increased on the ungreased steel wires, and the pitting corrosion had reached the steel substrate. For greased steel wires, the pitting corrosion only appeared in the zinc layer.

Corrosion losses on zinc plates and galvanized steel wires were compared. A higher loss was observed from the zinc plates and individual steel wires than those from the greased and ungreased cables. The results clearly showed that the presence of grease and the sheltering effect of the aluminum wires were factors in reducing corrosion on steel wires.

EDS and Raman spectroscopy analysis of the corroded steel wires, individual or extracted from the ACSR cables, revealed the corrosion products occurring in a saline atmosphere. This presence confirmed that the corrosion mechanism of these tests is not perturbed by an acceleration factor, although corrosion rates are very high compared to the ISO 12944-2 [15] corrosivity categories.

The tensile tests showed a significant reduction in the mechanical strength and deformation of individual wires from 168 days of corrosion, at the point when all the zinc layer is completely corroded. However, the galvanized steel wires from conductors did not show any mechanical loss for all corrosion durations. Fracture surfaces were observed and analyzed in this paper. These analyses revealed the evolution of the rupture surface for the corroded steel wires. For the uncorroded and low-corroded galvanized steel wire, the fracture surfaces were ductile, whereas high corrosion led to a more cleavage-like fracture.

**Author Contributions:** Conceptualization, A.R., L.G., L.D. and S.L.; methodology, A.R., L.G., L.D. and S.L.; investigation, A.R., L.G., L.D. and S.L.; writing—original draft preparation, A.R.; writing—review and editing, A.R., L.G., L.D. and S.L.; supervision, L.G., L.D. and S.L.; project administration, L.G., L.D. and S.L.; funding acquisition, L.G., L.D. and S.L. All authors have read and agreed to the published version of the manuscript.

**Funding:** This research was funded by LIA ECOMAT, Hydro-Québec, RTE, CRSNG, and InnovÉÉ.

**Data Availability Statement:** Data sharing does not apply to this article.

**Acknowledgments:** The authors and co-authors would like to acknowledge RTE and SolidAl for providing the cables and wires used in this study. The authors sincerely thank J. Creus from the

LASIE laboratory of the University of La Rochelle for providing access to the Raman spectroscopy in his Heritage Science Laboratory. The authors and co-authors would like to acknowledge Sahar Zouari from the SMC laboratory of the University Gustave Eiffel for helping with the EDS observations, and the technicians of the University of Sherbrooke and University Gustave Eiffel for their help, and expertise.

**Conflicts of Interest:** The authors declare no conflict of interest.

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
