# Peer review of "Degradation of Steel Wires in Bimetallic Aluminum–Steel Conductors Exposed to Severe Corrosion Conditions"

_cmd, doi:10.3390/cmd3040035_

Round 1

Reviewer 1 Report

The manuscript is devoted to the study of the degradation of steel wire on bimetallic aluminum-steel conductors in severe corrosion conditions. The authors convincingly showed that the presence of grease and the sheltering effect of aluminum wires are factors of reducing corrosion on steel wires. The manuscript is of theoretical and practical interest and can be published.

            However, the authors should take into account a number of minor remarks.

1. Lines 167-169. The authors should explain what is the difference between gamma, delta, zeta, eta intermetallic layers.

2. Figure 12. Inconsistency between the caption and the figure. Confused (c) and (e)

3. Lines 293-294:"These MEB and REMAN observations led to the same pollutant results for individual and cable steel  wires. " What is MEB and REMAN?

4. Figure 16. Authors should indicate the difference between (a) and (b).

5. Figure 17. Authors should indicate the difference between (a) and (b).

Reviewer 2 Report

This work implements an accelerated corrosion test in multiple samples relevant for ACSR conductors. The authors evaluated the corrosion attack and the mass loss of material after different exposure times. Corrosion products and mechanical properties were also assessed and correlated with the corrosion test results. In general, the manuscript is logical, and results are somewhat convincing. The novelty of this study and its benefit for other researchers is not properly highlighted. Some parts of the manuscript are difficult to follow, and the discussion of results must be improved. Please, carefully revise the following comments:

1. Consider the following paragraph (lines 67-71): “The impact of salt spray tests has been studied by N. LeBozec et al. [9] in comparison with other automotive corrosion tests. The ISO 9227 [7] has been studied in comparison with other automotive accelerated corrosion test standards, showing that the ISO 9227 test is the most severe corrosion test in terms of metal loss measured on cold-rolled steel panels”. This reviewer understands that LeBozec et al. compared the corrosion test from ISO 9227 with other automotive corrosion tests (not specified) and concluded that the one from ISO 9227 is the most severe. Please rephrase the paragraph to make this idea clear. This is an example on how the writing should be polished to help the reader easily understand the manuscript.

2. What is the contribution of this work? Please emphasize the novelty of this study in the manuscript.

3. Suppliers of all materials and equipment are not specified.

4. Quality of Figure 1 must be improved. Explain in the captions what the gray and white areas are.

5. The following statement does not add any information relevant to the obtained results: “Aluminum wires and plates were also introduced in the environmental chamber, but they will not be presented in this paper” (lines 98-100). Please consider removing it, further explaining why such decision was made, or adding those results on a supporting information section.

6. Figure 4 should have labels of the parts displayed (see Figure 2 as example).

7. The sections of the paper should be restructured. Results and Discussion section must be combined and added as number 3, and the current sections 3 to 5 as subsections of it.

8. There are a few optical images that are not in focus and thus, they don’t have the required quality for publication. Examples of the above are Figure 5, Figure 6 (336 days), and Figure 9. Please replace these images.

9. Consider adding labels of exposure time inside the optical images from Figures 6 to 8 to save space. Another alternative is labeling them as a), b), c), etc., and explaining in the corresponding figure caption what the exposure time is for each one.

10. In lines 186 to 189, the exposure time of 336 days is not mentioned in the analysis of the corrosion attack for the individual steel wire. Please check and modify the paragraph to include this exposure time.

11. In Figure 10, please extend the values in the X axis such as the standard deviation bar in the last point (at 336 days) is visible for all curves. Please increase the font size of both X and Y axis labels and values (see Fig. 3 as example).

12. The mass/thickness loss analysis is embedded in the Microstructural observations section. This analysis should be a separate section.

13. From lines 238-240: “According to the results obtained on ungreased 238 and greased cables, the estimation of total corrosion of galvanic layers on internal steel 239 wires is calculated at 726 days for ungreased wires, and 11,900 days (around 33 years) 240 for greased wires in this severe environment”. Please elaborate more and describe the calculations done for the estimation of total corrosion of galvanic layers.

14. In the EDS results section, please compare the same type of EDS data for both uncorroded and corroded wires. In the current version, the EDS spectra of the uncorroded wire is presented as graphs but without an image of where it was acquired (even when it is mentioned). For the corroded wire, the EDS data is shown as elemental mapping.

15. From lines 286-288: “Some RAMAN analyses were made by Antunes et al. [18] on carbon steel plates, revealing the presence of lepidocrocite (É£-FeO(OH)), and magnetite (Fe3O4) on the steel surface, after salt spray tests.” This statement mentions Fe corrosion products that are not discussed in the Raman results. Please remove it and add more discussion for the corrosion products that were found.

16. From lines 293-294: “These MEB and REMAN observations”. Please correct these terms to SEM and Raman.

17. Figure 13 caption is incomplete. Please indicate what corresponds to (a) and (b). Also, please consider changing the wavenumber for the actual compound.

18. In the X axis of Figure 15, the decimal separator symbol used is decimal comma, whereas the authors use decimal point throughout the text. Please also increase font size labels and values.

19. The Mechanical tension test section is hard to digest. Data must be organized better to help the reader understand the rationale and interpretation of results. The UTS value changes from test to test and thus does not give comparable information.

20. Discussions and Conclusion section must be changed to only Conclusions section. This is a follow-up on comment 7 above.

21. Please organize references with the same format.

22. It seems that are some references missing. Some examples can be find below:

§  Isozaki, M., Adachi, K., Hita, T. and Asano, Y. (2008), Study of corrosion resistance improvement by metallic coating for overhead transmission line conductor. Elect. Eng. Jpn., 163: 41-47. https://doi.org/10.1002/eej.20365

§  E. Håkansson, P. Predecki and M. S. Kumosa, Galvanic Corrosion of High Temperature Low Sag Aluminum Conductor Composite Core and Conventional Aluminum Conductor Steel Reinforced Overhead High Voltage Conductors, in IEEE Transactions on Reliability, vol. 64, no. 3, pp. 928-934, Sept. 2015, doi: 10.1109/TR.2015.2427894.

§  Ujah, C.O., Popoola, A.P.I. & Popoola, O.M. Review on materials applied in electric transmission conductors. J Mater Sci 57, 1581–1598 (2022). https://doi.org/10.1007/s10853-021-06681-9

Please check carefully again literature and add all the relevant references.

Round 2

Reviewer 2 Report

I have no further comments on this work. This reviewer recommends acceptance of the manuscript.